# OsOSCA1.1 Mediates Hyperosmolality and Salt Stress Sensing in *Oryza sativa*

**DOI:** 10.3390/biology11050678

**Published:** 2022-04-28

**Authors:** Yang Han, Yinxing Wang, Yuanjun Zhai, Zhaohong Wen, Jin Liu, Chao Xi, Heping Zhao, Yingdian Wang, Shengcheng Han

**Affiliations:** 1Beijing Key Laboratory of Gene Resources and Molecular Development, College of Life Sciences, Beijing Normal University, Beijing 100875, China; 201431200013@mail.bnu.edu.cn (Y.H.); 202021200021@mail.bnu.edu.cn (Y.W.); 201731200012@mail.bnu.edu.cn (Y.Z.); 201231200013@mail.bnu.edu.cn (Z.W.); liujin@bnu.edu.cn (J.L.); xichao@bnu.edu.cn (C.X.); hpzhao@bnu.edu.cn (H.Z.); ydwang@bnu.edu.cn (Y.W.); 2Academy of Plateau Science and Sustainability of the People’s Government of Qinghai Province & Beijing Normal University, Qinghai Normal University, Xining 810008, China

**Keywords:** OsOSCA, hyperosmolality-induced calcium increases, salt stress-induced calcium increases, OsOSCA1.1-regulated gene expression, rice

## Abstract

**Simple Summary:**

Drought and salt stresses are the two major abiotic constraints on plant growth and development, as well as crop production. Previous studies have shown that several of 11 OsOSCAs complement hyperosmolality-induced [Ca^2+^]_cyt_increases (OICI_cyt_), salt stress-induced [Ca^2+^]_cyt_ increases (SICI_cyt_), and the associated growth phenotype in the *Arabidopsis* mutant *osca1*. However, their biological functions in rice are still unclear. In this paper, we found that OsOSCA1.1 mediates OICI_cyt_ and SICI_cyt_ in rice roots, which are associated with stomatal closure and seedling survival, in response to hyperosmolality and salt stress. The transcriptomic analysis revealed the following three types of OsOSCA1.1-regulated genes in shoots: 2416 sorbitol, 2349 NaCl, and 1844 osmotic-stress response genes. The GO enrichment analysis showed that these OsOSCA1.1-regulated genes were enriched in transcription regulation, hormone responses, and phosphorylation terms, consistent with the enrichment of cis-regulatory elements, i.e., ABRE, ARE, MYB, and MYC binding elements, in the 2000-bp promoter regions of these genes. Our results provide important clues for dissecting calcium regulated gene expression networks in response to drought and salt stresses in rice.

**Abstract:**

OSCA (*reduced hyperosmolality-induced [Ca^2+^]_i_ increase*) is a family of mechanosensitive calcium-permeable channels that play a role in osmosensing and stomatal immunity in plants. *Oryza sativa* has 11 *OsOSCA* genes; some of these were shown to complement hyperosmolality-induced [Ca^2+^]_cyt_ increases (OICI_cyt_), salt stress-induced [Ca^2+^]_cyt_ increases (SICI_cyt_), and the associated growth phenotype in the *Arabidopsis thaliana* mutant *osca1*. However, their biological functions in rice remain unclear. In this paper, we found that OsOSCA1.1 mediates OICI_cyt_ and SICI_cyt_ in rice roots, which are critical for stomatal closure, plant survival, and gene expression in shoots, in response to hyperosmolality and the salt stress treatment of roots. Compared with wild-type (Zhonghua11, ZH11) plants, OICI_cyt_ and SICI_cyt_ were abolished in the roots of 10-day-old *ososca1.1* seedlings, in response to treatment with 250 mM of sorbitol and 100 mM of NaCl, respectively. Moreover, hyperosmolality- and salt stress-induced stomatal closure were also disrupted in a 30-day-old *ososca1.1* mutant, resulting in lower stomatal resistance and survival rates than that in ZH11. However, overexpression of *OsOSCA1.1* in *ososca1.1* complemented stomatal movement and survival, in response to hyperosmolality and salt stress. The transcriptomic analysis further revealed the following three types of OsOSCA1.1-regulated genes in the shoots: 2416 sorbitol-responsive, 2349 NaCl-responsive and 1844 common osmotic stress-responsive genes after treated with 250 mM of sorbitol and 125 mM NaCl of in 30-day-old rice roots for 24 h. The Gene Ontology enrichment analysis showed that these OsOSCA1.1-regulated genes were relatively enriched in transcription regulation, hormone response, and phosphorylation terms of the biological processes category, which is consistent with the *Cis*-regulatory elements ABRE, ARE, MYB and MYC binding motifs that were overrepresented in 2000-bp promoter regions of these OsOSCA1.1-regulated genes. These results indicate that OsOSCA-mediated calcium signaling specifically regulates gene expression, in response to drought and salt stress in rice.

## 1. Introduction

As sessile organisms, plants are exposed to continual abiotic stresses, including drought and salt stress, throughout their growth period, i.e., from seed germination to flowering [1,2]. The first responses to hyperosmolality and salt stress are rapid cytosolic [Ca^2+^] increases in plants [3]. Then, plants enable multiple signaling cascades that regulate the transcription of a specific set of genes to modulate the metabolism and physiology, to finally adapt to drought and salt stress [4,5]. Besides the common hyperosmotic phase for drought and salt stresses, salt stress also exerts ionic effects in cells and exhibits ion toxicity [2]. Thus, drought and salt stresses have both distinct and overlapping features [2]. Previous extensive studies have applied microarray, expressed sequence tag (EST), and RNA sequencing (RNA-seq) analysis techniques to identify drought- and salt stress-responsive genes in plants, including *Arabidopsis*, rice and barley plants [6,7,8]. These key genes are transcriptionally regulated by several families of transcription factors (TFs), including ABA-responsive element-binding proteins (AREB)/ABRE binding factors (ABF), dehydration-responsive element-binding proteins (DREB), ethylene-responsive factors (ERF), no apical meristem (NAM), ATAF1/2, and cup-shaped cotyledon (CUC2) (NAC) TFs, and basic leucine zipper (bZIP) family proteins MYC and MYB [8,9,10]. However, the molecular mechanism that allows plants to distinguish drought and salt stress signals, especially calcium signaling involved in regulating gene expression in response to drought and salt stresses, remains unknown.

Previous studies have demonstrated that *Arabidopsis OSCA1.1* encodes a hyperosmolality-gated calcium-permeable channel [11], and that OSCA1.2/calcium permeable stress-gated cation channel 1 (AtCSC1) can be activated by hyperosmotic stress, via 250 mM of NaCl and 500 mM of mannitol, in *AtCSC1*-expressing oocytes [12]. Recently, the protein structures of OSCA1.1, OSCA1.2, and OSCA3.1 were characterized by cryo-electron microscopy, which revealed that the OSCA family belongs to a new group of mechanosensitive calcium-permeable channels, with structural similarities to mammalian TMEM16-family proteins [13,14,15]. The 11 *OsOSCAs* in the rice genome [16] and the overexpression of *OsOSCA1.1*, OsOSCA1.4, or *OsOSCA2.2* in the *Arabidopsis* mutant *osca1* complemented OICI_cyt_ and SICI_cyt_, and influenced stomatal closure and root growth, in response to hyperosmolality salt stress treatment, and drought-related leaf water loss [17,18]. However, the biological functions of OsOSCAs in rice are still unknown. In this study, we investigated the role of OsOSCA1.1 on mediating OICI_cyt_ and SICI_cyt_ in rice roots that are associated with stomatal movement, in response to hyperosmolality and salt stress treatment. We also performed a transcriptomic analysis of rice shoots following the treatment with sorbitol and NaCl in roots to investigate the roles of OsOSCA1.1 regulating hyperosmolality-, ionic-, and osmotic-responsive genes. The findings of this study improve our understanding of the molecular mechanisms by which OsOSCAs sense various abiotic stresses to regulate gene expression in rice.

## 2. Materials and Methods

### 2.1. Plant Growth

Rice plants (*Oryza sativa* L. spp. *japonica* cv. Zhonghua11, ZH11) were planted, grown, and treated with hyperosmolality or salt stress as described [16]. One T-DNA insertion line RMD_04Z11JO28 (genetic background ZH11), named *ososca1.1*, was obtained from Huazhong Agricultural University [19] and confirmed by genomic polymerase chain reaction (PCR; Appendix A). Rice transformation was performed by Wuhan Biorun Bio-Tech Co., Ltd. (Wuhan, China), and generated T3-generation single-insertion homozygous transgenic rice lines. The RNA levels of *OsOSCA1.1* in the transgenic *ososca1.1* lines overexpressing *OsOSCA1.1* were determined by semi-quantitative PCR (Appendix A). The primers used are listed in Appendix A.

Rice seeds were sterilized with 2.5% NaClO for 30 min, rinsed with ddH_2_O six times, soaked in water at 20 °C for 3 days and germinated for 1 day at 37 °C. Then, the germinated seeds were transferred into bottomless 96-well plates, floated on water in a growth chamber with a 14 h light (24 °C)/10-h dark (20 °C) photoperiod. After five days, the seedlings were transferred into Yoshida’s culture medium, which was replaced every 2 days, until Ca^2+^ imaging and observation of phenotypes in response to different osmotic-related stress treatments.

### 2.2. Cameleon-Based Measurements of [Ca^2+^]_cyt_ in Rice Root Cells

A Förster resonance energy transfer (FRET)-based Ca^2+^ sensor, Yellow Cameleon 3.6 (YC3.6), was transferred into ZH11 and *ososca1.1* using *Agrobacterium tumefaciens* GV3101, generating three single-insertion homozygous lines. Approximately 5-mm hair roots were detached from 10-day-old seedlings and stabilized on glass slides, using low-melting agarose in an imaging buffer (10 mM MES, 5 mM KCl, 2.5 mM CaCl_2_; pH 5.8, adjusted with Tris base). FRET signals were monitored using an Oberserver-A1 microscope (Zeiss, Oberkochen, Germany) equipped with an arc lamp (Lambda DG4; Sutter Instruments, Novato, CA, USA) and an Optosplit II Image Splitter (Cairn Research Ltd., Sittingbourne, UK). Cyan fluorescent protein (CFP; 428.9 ± 5.5_Ex_/465 ± 32_Em_), yellow fluorescent protein (YFP; 502.6 ± 11.2_Ex_/549 ± 21_Em_), and FRET_raw_ (428.9 ± 5.5_Ex_/549 ± 21_Em_) images were captured every 3 s at room temperature. The apparent FRET efficiency (E_app_) was evaluated as described previously [3]. All fluorescence images were collected and processed using MetaFluor software. The data were analyzed using Matlab R2012b software (MathWorks, Natick, MA, USA) and plotted using Prism 5 software (GraphPad Software, Inc., San Diego, CA, USA).

### 2.3. Stomatal Aperture and Resistance Bioassays

The rice stomatal types were defined and analyzed according to the previous report with some modification [20]. The 30-day-old rice seedlings were transferred to Yoshida’s culture solution, containing 150 mM of sorbitol or 100 mM of NaCl for 0, 1, or 36 h. The leaves were not allowed to come into contact with the culture solution, and were then sampled for imaging and stomatal resistance measurements. For stomata imaging, the leaves were detached and immediately fixed in liquid nitrogen, and images were acquired using an environmental scanning electron microscope (Quanta 200; FEI, Brno, Czech Republic). The width and length of the stomatal apertures were analyzed using DP2-BAW software (Olympus Corp., Tokyo, Japan). To measure the leaf stomatal resistance, new grown and fully expanded leaves were detected using a transient-time porometer (AP4; Delta-T Devices, Cambridge, UK) at 50–60% relative humidity, under a constant temperature of 28 °C.

### 2.4. Osmotic Stress-Related Survival Assay

The twenty-day-old rice seedlings were transferred to Yoshida’s culture solution containing 20% polyethyleneglycol (PEG) 6000 or 150 mM of NaCl for 10 days, then the seedlings were recovered in Yoshida’s culture solution for 18 days. The seedlings were photographed at 7 days of the stress treatment phase, and after 18 days of the recovery phase; and the seedling survival rate was determined after 18 days of the recovery phase.

### 2.5. RNA Extraction and Sequencing Analysis

We treated 30-day-old ZH11 and *ososca1.1* plants with 250 mM of sorbitol or 125 mM of NaCl in Yoshida’s culture solution for 0 and 24 h, respectively. The total RNA was extracted from the shoots using TRIzol reagent (Invitrogen, Carlsbad, CA, USA) and purified by removing DNA using the PureLink DNase Kit and PureLink RNA Mini Kit (Invitrogen). The libraries were generated using the NEBNext Ultra RNA Library Prep Kit for Illumina (NEB, Ipswich, MA, USA) according to the manufacturer’s instructions and sequenced on the Illumina HiSeq 2000 platform (Illumina, San Diego, CA, USA) at Novogene Bioinformatics Technology Co., Ltd. (Beijing, China). After sequencing, the reads were cleaned by removing the adaptor and low-quality reads (bases of Q_phred_ ≤ 20), and were then mapped and aligned with the rice reference genome (genome (http://rice.plantbiology.msu.edu/, accessed on 20 October 2019). Next, a gene index was generated using the Bowtie v.2.0.6 alignment tool (Broad Institute, Cambridge, MA, USA), according to gene model annotation files downloaded from the rice genome website (ftp://ftp.ensemblgenomes.org/pub/release-23/plants/fasta/oryza_sativa/dna/, accessed on 20 October 2019). The value of the fragments per kilobase of exon per million mapped fragments (FPKM) for each gene was used to assess the expression level. The differentially expressed genes (DEGs) between the two groups were identified using the DEGSeq package [21] and genes with a fold change of ≥2 and adjusted *p* < 0.05 were considered as DEGs. The unique and shared DEGs expressed in rice shoots were classified using Venn diagrams (http://bioinformatics.psb.ugent.be/software/details/Venn-Diagrams, accessed on 20 October 2019). In addition, hierarchical clustering was performed using Cluster 3.0 software (http://bonsai.hgc.jp/~mdehoon/software/cluster/, accessed on 10 October 2021) and a heat map was generated using the Java TreeView-1.1.6r4-win visualization tool (https://sourceforge.net/projects/jtreeview/files/jtreeview/, accessed on 4 November 2021). Log transformation and mean centering were applied, and the Euclidean distance was calculated.

### 2.6. Gene Ontology (GO) Enrichment Analysis

GO term information, which included biological process, cellular component, and molecular function categories for DEGs, was extracted from the BLAST results against the SWISS-PROT database. The GO enrichment analysis was conducted using Plant Slim and BinGO v2.44 [22]. Any GO term with a false discovery rate (FDR) <0.01 was considered significantly overrepresented in the test gene set.

### 2.7. Cis-Acting Regulatory Element Analysis

The consensus *cis*-regulatory elements within 2000 bp of sequences upstream of the ATG site of *OsOSCA1.1*-regulated genes were predicted using the PlantCARE web tool (https://bioinformatics.psb.ugent.be/webtools/plantcare/html/, accessed on 5 February 2022). 

### 2.8. Statistical Analyses

Statistical analyses were performed using data processing system software [23]. One-way analysis of variance (ANOVA) and Tukey’s multiple range test were used to assess the significant differences, where *p* < 0.05 and *p* < 0.01 indicate significant and highly significant differences, respectively.

## 3. Results

### 3.1. OsOSCA1.1 Is Required for OICI_cyt_ and SICI_cyt_ in Rice Roots

To determine whether OsOSCA1.1 mediates OICI_cyt_ and/or SICI_cyt_ in rice, we used YC3.6 to assess the change of [Ca^2+^]_cyt_ levels in the root cells of 10-day-old ZH11 and *ososca1.1* plants in response to hyperosmolality and salt stress, respectively. Following the treatment with 250 mM of sorbitol, a rapid ~99% increase in the FRET ratio (relative E_app_) occurred in the root cells of ZH11, whereas the increase was only ~10% in *ososca1.1*. Thus, OICI_cyt_ was greatly disrupted in *ososca1.1* (Figure 1A–C). The increase in FRET ratio was ~66% in the roots of ZH11, but only ~7.5% in those of *ososca1.1* in response to 100 mM of NaCl (Figure 1D–F), suggesting that SICI_cyt_ was also despaired in the root cells of *ososca1.1*. These results demonstrate that OsOSCA1.1 is essential for OICI_cyt_ and SICI_cyt_ in rice root cells. 

### 3.2. OsOSCA1.1 Mediates Hyperosmolality- and Salt Stress-Induced Stomatal Closure

To elucidate the physiological role of OsOSCA1.1 in response to hyperosmolality and salt stress at the whole-plant level, we monitored the stomatal status of 30-day-old seedlings of ZH11, *ososca1.1*, and two independent *OsOSCA1.1*-complemented lines (*OsOSCA1.1^OE^* in *ososca1.1* lines 1 and 2), in response to hyperosmolality and salt stress. Compared to ZH11, both hyperosmolality-induced stomatal closure (HISC) and salt-induced stomatal closure (SISC) were abolished in *ososca1.1* following treatment with 150 mM of sorbitol and 100 mM of NaCl for 1 and 36 h, respectively (Figure 2A,B). HISC and SISC impairment was reversed in *OsOSCA1.1^OE^* in *ososca1.1* lines 1 and 2 (Figure 2A,B). We further measured the stomatal resistance in ZH11, *ososca1.1*, and *OsOSCA1.1^OE^* in *ososca1.1* lines 1 and 2, treated with 150 mM of sorbitol and 100 mM of NaCl for 1 and 36 h, respectively, and obtained similar results using a stomatal aperture assay (Figure 2C). In addition, no significant difference in stomatal density was detected among ZH11, *ososca1.1*, and *OsOSCA1.1^OE^* in *ososca1.1* lines 1 and 2 (Appendix A), indicating that the changes in stomatal resistance in response to hyperosmolality and salt stress were mainly due to stomatal closure.

Next, we investigated the survival rates of ZH11, *ososca1.1*, and *OsOSCA1.1^OE^* in *ososca1.1* lines 1 and 2 in response to hyperosmolality and salt stress. Following treatment with 20% PEG for 7 days, the leaves exhibited more wilting. Moreover, shorter shoots were observed in *ososca1.1* than ZH11, and leaf chlorosis was more severe in *ososca1.1* than ZH11, following treatment with 150 mM of NaCl for 7 days (Figure 2D). After 3 more days of 20% PEG and 150 mM of NaCl treatment, the plants were transferred to fresh Yoshida’s culture solution, for recovery for 18 days. Virtually all ZH11 individuals, but only half of the *ososca1.1* individuals, survived the 20% PEG treatment (Figure 2E). Following treatment with 150 mM of NaCl, the survival rate of *ososca1.1* was ~2.78%, which was substantially lower than that of ZH11 (~44.58%; Figure 2E). The lower survival rate of *ososca1.1,* in response to hyperosmolality and salt stress, was reversed in *OsOSCA1.1^OE^* in *ososca1.1* lines 1 and 2 (Figure 2D,E). These results suggest that OsOSCA1.1 mediates OICI and SICI in root cells, which is related to HISC and SISC, and survival in response to hyperosmolality and salt stress. 

### 3.3. OsOSCA1.1-Regulated Gene Expression in Response to Hyperosmolality and Salt Stress

OsOSCA1.1 mediation of OICI and SICI enables the assessment of calcium-regulated hyperosmolality and salt-stress responsive gene expression networks in rice. Therefore, we investigated the gene expression profiles of shoots from 30-day-old ZH11 and *ososca1.1* plants treated with 250 mM of sorbitol or 125 mM of NaCl for 24 h. The RNA samples were analyzed using the HiSeq 2000 platform; after removing the primer adaptor sequences and low-quality reads, approximately 40–47 million high-quality reads were generated for each sample (Appendix A). Then, over 91% of the genes with at least one read among the sample were mapped in the rice genome (~30,745 coding sequences; Appendix A), suggesting that each sample had sufficient sequence depth to analyze gene expression in response to hyperosmolality and salt stress.

The DEG analysis was based on the FPKM; genes with a two-fold change and that adjusted *p* < 0.05 between the two samples were classified as DEGs, as stated above. A total of 587 DEGs were first identified between ZH11 and *ososca1.1*. Then, we identified 4282 hyperosmolality response genes in ZH11, including 302 of 587 DEGs between ZH11 and *ososca1.1*, and 5206 hyperosmolality response genes in *ososca1.1,* including 377 DEGs between ZH11 and *ososca1.1* (Figure 3A and Appendix A). In addition, 3916 and 3823 salt stress response genes were identified in ZH11 and *ososca1.1*, including 304 and 300 DEGs between ZH11 and *ososca1.1*, respectively (Figure 3B and Appendix A). Moreover, based on the expression levels and patterns of each DEG, we further identified 6609 OsOSCA1.1-regulated genes, which were classified into the following three groups: 2416 sorbitol, 2349 NaCl, and 1844 osmotic stress response genes (Figure 3C).

A GO annotation enrichment analysis was further performed to characterize these three groups of OsOSCA1.1-regulated genes. In the “biological process” category, the transcripts were commonly enriched in “responses to stimuli” term, including “response to hormones, wounding, lipids, alcohol, oxygen-containing compounds, and light” terms, and “protein phosphorylation” term was also enriched among all three groups of the OsOSCA1.1-regulated genes (Figure 3D–F). The transcripts of OsOSCA1.1-regulated sorbitol and NaCl response genes were also enriched in “transcription regulation” term (Figure 3D,F), and those of osmotic-stress response genes were specifically enriched in “cell division” term (Figure 3E). 

Moreover, in the “cellular component” category, the transcripts were enriched in “cellular anatomical entity” term, including “cell periphery”, “membrane”, “plasma membrane”, “intrinsic component of membrane”, and “integral component of membrane” terms, among all three of the OsOSCA1.1-regulated genes (Appendix A). In addition, the transcripts of the OsOSCA1.1-regulated osmotic-stress response genes were specifically enriched in “microtubule”, “apoplast”, and “nuclear replisome” terms, and those of the NaCl response genes specifically in “photosystem I”, “photosystem II”, “nucleous”, and “chromosomal region” terms (Appendix A). 

In addition, in the “molecular function” category, the transcripts were enriched in “oxidoreductase activity”, “protein kinase activity”, and “ATP binding” terms among all three of the OsOSCA1.1-regulated genes (Appendix A). The transcripts of the OsOSCA1.1-regulated sorbitol and NaCl response genes were enriched in “DNA-binding transcription factor activity” term, and those of the sorbitol and osmotic-stress response genes were enriched in “metal ion binding” term (Appendix A). However, sorbitol response genes enriched in “calcium binding” term, but osmotic-stress response genes in “iron ion binding” and “manganese ion binding” terms (Appendix A). In addition, the transcripts of the OsOSCA1.1-regulated sorbitol response genes were specifically enriched in “chitinase activity”, and “carbohydrate binding”, those of the osmotic-stress response genes specifically in “microtubule motor activity”, “single-stranded DNA binding” and “single-stranded DNA helicase activity”, and those of NaCl response genes specifically in “UDP-glycosyltransferase”, “hexosyltransferase activity”, and “heme binding” terms (Appendix A). Thus, we concluded that OsOSCA1.1 regulates gene expression via both common and unique pathways, in response to hyperosmolality and salt stress in rice.

### 3.4. Identification of OsOSCA1.1-Regulated Kinases, Transcription Factors and Hormone Components in Response to Hyperosmolality and Salt Stress

The GO enrichment analysis showed that the OsOSCA1.1-regulated genes were mainly enriched in hormone responses, phosphorylation, and transcription regulation (Figure 3). Therefore, we removed the gene redundancies from these three terms and identified 22 kinases, including calcineurin B-like protein-interacting protein kinases (CIPKs) and calcium-dependent protein kinases (CPKs), as OsOSCA1.1-regulated osmotic stress and NaCl response genes. We also found that 49 TFs, including MYB, NAC, and WRKY family members, belonged to the OsOSCA1.1-regulated sorbitol- and NaCl response genes. Many hormone components differentially expressed in response to hyperosmolality and salt stress were also identified (Figure 4 and Appendix A). Among these hormone components, auxin, brassinosteroids (BRs), and ethylene components were included in all three OsOSCA1.1-regulated gene types, while cytokinin, gibberellic acid (GA) and jasmonic acid (JA) components were included in the OsOSCA1.1-regulated sorbitol and NaCl response genes. ABA components were excluded from the OsOSCA1.1-regulated NaCl response genes (Figure 4). These results indicate the existence of a complex transcription regulatory network, in which OsOSCA1.1 regulates gene expression for the adaption to hyperosmolality and salt stress in rice.

### 3.5. ABRE, ARE, MYB and bHLH Binding Sites Are Extensively Distributed in the Promoters of OsOSCA1.1-Regulated Genes

To identify the potential OsOSCA1.1-activated DNA cis-regulatory sites, we first hierarchically clustered these three groups of OsOSCA1.1-regulated genes, according to their FPKM values. Among 2416 sorbitol response genes, we selected 119 amalgamated OsOSCA1.1-downregulated and 108 OsOSCA1.1-upregulated genes for clustering (Figure 5A). Among 1844 osmotic-stress response genes, 70 OsOSCA1.1-upregulated and 56 OsOSCA1.1-downregulated genes were selected for clustering (Figure 5B). Among 2349 NaCl response genes, we selected 82 NaCl-upregulated and 79 NaCl-downregulated genes that showed OsOSCA1.1 downregulation (Figure 5C). Next, we entered the 2000-bp promoter regions of these genes into the PlantCARE online tool to identify the overrepresented motifs. Firstly, two motifs, C/TAACNA/G and CANNTG, which are separate MYB and bHLH TF binding sites, were widely distributed in the promoters of nearly all of the OsOSCA1.1-regulated genes described above (Figure 5D). The third motif, ACGT, which is the core sequence of the G-box or ABA-responsive element, and the fourth motif, ARE, which is a *cis*-acting regulatory element essential for anaerobic induction, were identified in more than 80% of the OsOSCA1.1-regulated gene promoters (Figure 5D). In addition, a drought response element (DRE) and low-temperature response element (LTR), were identified in more than 50% of the OsOSCA1.1-regulated gene promoters (Figure 5D). The AuxRR-core and TGA-element motifs, and the P-box and TATC-box motifs, are *cis*-acting regulatory elements involved in auxin and GA responses, respectively, and were identified in less than 50% of theOsOSCA1.1-regulated gene promoters (Figure 5D). These results are consistent with the OsOSCA1.1-regulated gene enrichment in “stimulus response”, “transcription regulation” and “DNA-bind transcription factor activity” GO terms.

## 4. Discussion

Previous studies have shown that a variety of species possess multiple members of the OSCA gene family; for example, 15 orthologs are found in *Arabidopsis* [11], 11 in rice [16], 12 in maize [24], 4 in yeast, and 3 in vertebrates [12]. We further showed that OsOSCA1.4 is a plasma membrane-localized ion channel that mediates OICI_cyt_ and SICI_cyt_ in HEK293 cells and *Arabidopsis osca1* plants [17]. Although OsOSCA1.1 and OsOSCA2.2 were mainly localized in the endoplasmic reticulum membrane in *Arabidopsis* mesophyll protoplasts, overexpression of *OsOSCA1.1* or *OsOSCA2.2* in *osca1* complemented OICI_cyt_ and SICI_cyt_, as well as the associated growth phenotype in response to hyperosmolality and salt stress [18]. In this paper, we characterized the function of OsOSCA1.1 in rice, and first found that OsOSCA1.1 mediates OICI_cyt_ and SICI_cyt_ in rice roots, which are associated with stomatal closure and seedling survival in response to hyperosmolality and salt stress treatment. Then, we performed a transcriptomic analysis in ZH11 and *ososca1.1* to dissect the calcium regulated gene expression networks in response to hyperosmolality and salt stress in rice. Therefore, we identified the following three types of OsOSCA1.1-regulated genes: sorbitol, NaCl and osmotic stress response genes, which were enriched in the unique and shared GO terms, indicating that rice plants present distinct and overlapping feature in gene expression profiles in response to drought and salt stresses. 

In plants, drought and salt stresses can up- and down-regulate gene expression in response to these stimuli [25]. A previous study identified 5284 DEGs in rice leaf, root, and young panicle under drought stress [26], and another study identified 1563 upregulated and 1746 downregulated genes in the drought-susceptible rice variety IR64 under water-deficit stress [27]. Using RNA-seq, 5273 DEGs were identified between the salt-tolerant and -sensitive genotypes of Indica rice at the seedling stage [28]. The functional annotation of these DEGs showed that most encoded multiple members of different signaling networks, including hormone- and calcium-signaling pathways, mitogen-activated protein kinase (MAPK) cascades, and TFs, indicating that these genes play important roles in the drought and salt stress responses of rice [28]. However, the molecular networks, regulated by calcium signaling pathways in response to drought and salt stress in plants, remain unclear. The application of electrical stimulation to *Arabidopsis* seedlings expressing cytosolic aequorin produced the following three distinct [Ca^2+^]_cyt_ elevation levels: single transients [29], resembling responses to osmotic stress and salt stress [3]; prolonged elevation, as previously reported in response to slow cooling [30]; and oscillations, as previously reported in guard cells responding to ABA, and in legume root hairs responding to Nod factors [31,32]. A total of 269 [Ca^2+^]_cyt_-upregulated genes were identified in *Arabidopsis* seedlings, through full-genome microarray analysis [29]. In the present study, we showed that OICI_cyt_ and SICI_cyt_ in rice root cells present a single transient elevation. Because *ososca1.1* exhibited both OICI_cyt_ and SICI_cyt_ disruptions in rice root cells, we characterized 6609 OsOSCA1.1-regulated genes in rice shoots using RNA-seq analyses, and these genes were classified into the following three groups: 2416 sorbitol, 2349 NaCl, and 1844 osmotic stress response genes. The GO enrichment analysis further showed that these OsOSCA1.1-regulated genes are enriched in “transcription regulation”, “hormone responses”, and “phosphorylation” terms. These results suggested that OsOSCA1.1-mediated calcium signals crosstalk with hormone signaling pathways, for regulating gene expression in response to hyperosmolality and salt stress in rice. In addition, it is interesting that, besides the common “cellular anatomical entity” term of the “cellular component” category in these three groups of OsOSCA1.1-regulated genes, the transcripts of the OsOSCA1.1-regulated osmotic-stress response genes were enriched in “microtubule” and “nuclear replisome” terms, which indicated that the common osmotic phase of drought and salt stress affect the microtube-dependent DNA replication and division. However, the NaCl response genes were specifically enriched in “photosystem I” and “photosystem II” terms, suggesting that sodium ion from salt stress regulates the expression of photosystem genes. 

A previous study showed that four [Ca^2+^]_cyt_-regulated promoter motifs, i.e., ABRE, DRE/CRT element, CAM box (CGCG box, binding site of the CAMTA TFs) [33], and site II box (binding motif of the TCP TFs) [34], have been characterized in *Arabidopsis* [29]. Similarly, using the PlantCARE program, we also found that the ABRE and DRE motifs were extensively enriched in the promoters of the OsOSCA1.1-regulated genes. In addition, the MYB TF binding site C/TAACNA/G and bHLH TF binding site CANNTG, which are involved in hormone signaling crosstalk, including ABA, JA, BR, SA, and ethylene [35,36], were overrepresented in the promoters of the OsOSCA1.1-regulated genes, indicating that these two motifs are also [Ca^2+^]_cyt_-regulated motifs. In addition, previous studies showed that the ARE motif is a cis-acting regulatory element essential for anaerobic induction [37]; LTR is enriched in the genes responding to drought, low temperature, or high salt [38,39]; the AuxRR-core, TGA-element, P-box and TATC-box motifs are *cis*-acting regulatory elements involved in auxin and GA responsiveness [40]. These motifs were also distributed in the promoter regions of most OsOSCA1.1-regulated genes. These results suggested that OsOSCA1.1-mediated calcium signaling is required for plant sensing and adaption to environmental stress in rice.

## 5. Conclusions

In this paper, we found that OsOSCA1.1 mediates OICI_cyt_ and SICI_cyt_ in rice roots, which are pivotal for stomatal closure and seedling survival in response to hyperosmolality and salt stress. A transcriptomic analysis following the treatment of 30-day-old rice roots with 250 mM of sorbitol and 125 mM of NaCl for 24 h revealed the following three types of OsOSCA1.1-regulated genes in shoots: sorbitol response, NaCl response, and common osmotic stress response genes. A GO enrichment analysis showed that these OsOSCA1.1-regulated genes were enriched in transcription regulation, hormone responses, and phosphorylation, consistent with the enrichment of *cis*-regulatory elements, i.e., ABRE, ARE, MYB, and MYC binding elements, in the 2000-bp promoter regions of these OsOSCA1.1-regulated genes. This study provides valuable clues for the functional characterization of OsOSCA1.1-mediated calcium signaling in rice.

## Figures and Tables

**Figure 1 biology-11-00678-f001:**
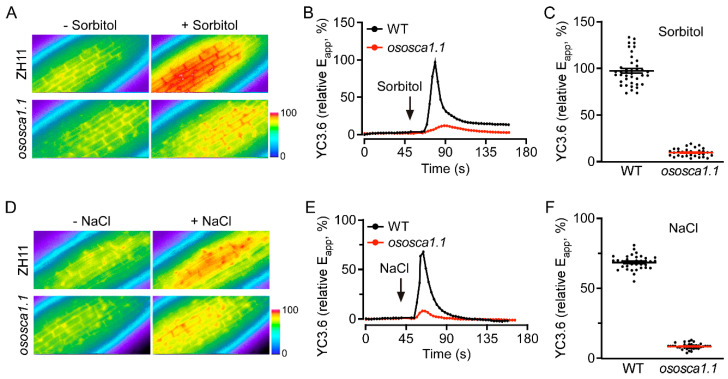
OsOSCA1.1 mediates hyperosmolality-induced [Ca^2+^]_cyt_ (OICI_cyt_) and salt-induced [Ca^2+^]_cyt_ (SICI_cyt_) increases in rice seedling roots. (**A**–**C**) OICI_cyt_ was monitored in the root cells of ZH11 and *ososca1.1,* expressing Yellow Cameleon YC3.6 (YC3.6) under treatment with 250 mM of sorbitol. E_app_, relative apparent Förster resonance energy transfer (FRET) efficiency of YC3.6. (**A**) Typical image before and after treatment, (**B**) changes in [Ca^2+^]_cyt_ over time, and (**C**) mean change in [Ca^2+^]_cyt_ in response to 250 mM of sorbitol in ZH11 and *ososca1.1* [means ± standard deviation (SD); *n* = 40]. In (**B**), black and red lines indicate [Ca^2+^]_cyt_ in ZH11 and *ososca1.1*, respectively; black arrow indicates hyperosmolality stress initiation. In (**C**), black dots indicate individual experiments; black and red lines indicate mean changes. (**D**–**F**) SICI_cyt_ was analyzed in the root cells of ZH11 and *ososca1.1* expressing YC3.6 under treatment with 100 mM of NaCl. (**D**) Typical image before and after treatment, (**E**) changes in [Ca^2+^]_cyt_ over time, and (**F**) mean change in [Ca^2+^]_cyt_ in response to 100 mM NaCl in ZH11 and *ososca1.1* (means ± SD; *n* = 40). In (**E**), black and red lines indicate [Ca^2+^]_cyt_ in ZH11 and *ososca1.1*, respectively; black arrow indicates salt stress initiation. In (**F**), black dots indicate individual experiments; black and red lines indicate mean changes.

**Figure 2 biology-11-00678-f002:**
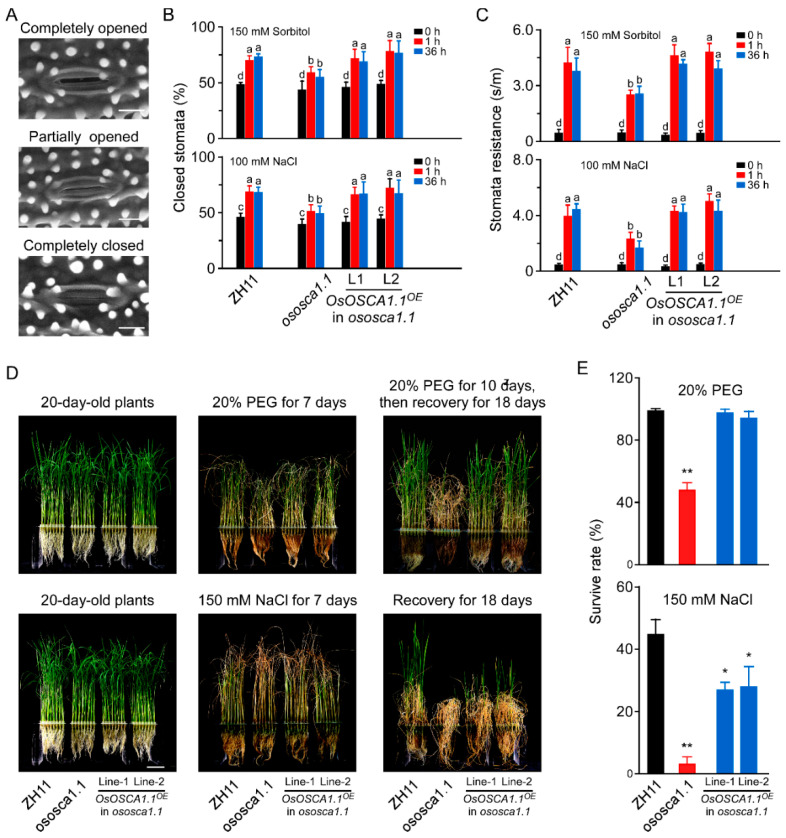
OsOSCA1.1 is essential for hyperosmolality- and salt-induced stomatal closure and survival under hyperosmolality and salt stress conditions in rice. (**A**) Environmental scanning electron micrographs of opening/closing stomata on a 30-day-old rice blade. Scale bar = 10 μm. (**B**) Percentage of closed stomata in 30-day-old ZH11, *ososca1.1*, and *OsOSCA1.1-*overexpressing *ososca1.1* line 1 and 2 plants treated with 150 mM of sorbitol and 100 mM of NaCl for 1 and 36 h, respectively (*n* = 130 stomata). Different letters indicate significant differences at *p* < 0.05 [one-way analysis of variance (ANOVA)]. (**C**) Stomatal resistance of 30-day-old ZH11, *ososca1.1*, and *OsOSCA1.1-*overexpressing *ososca1.1* lines 1 and 6 treated with 150 mM of sorbitol and 100 mM of NaCl for 1 and 36 h, respectively (*n* = 12 leaves). ^a,b,c,d^ Different letters indicate significant differences at *p* < 0.05 (one-way ANOVA). (**D**) Images of 20-day-old ZH11, *ososca1.1*, and *OsOSCA1.1-*overexpressing *ososca1.1* lines 1 and 2 treated with 20% polyethyleneglycol (PEG) or 150 mM of NaCl for 7 days, followed by 3 more days of treatment and recovery for 18 days. (**E**) Survival rate of ZH11, *ososca1.1*, and *OsOSCA1.1-*overexpressing *ososca1.1* lines 1 and 2 treated with 20% PEG or 150 mM NaCl for 10 days, followed by recovery for 18 days (*n* = 300 seedlings). * *p* < 0.05; ** *p* < 0.01 (Tukey’s test).

**Figure 3 biology-11-00678-f003:**
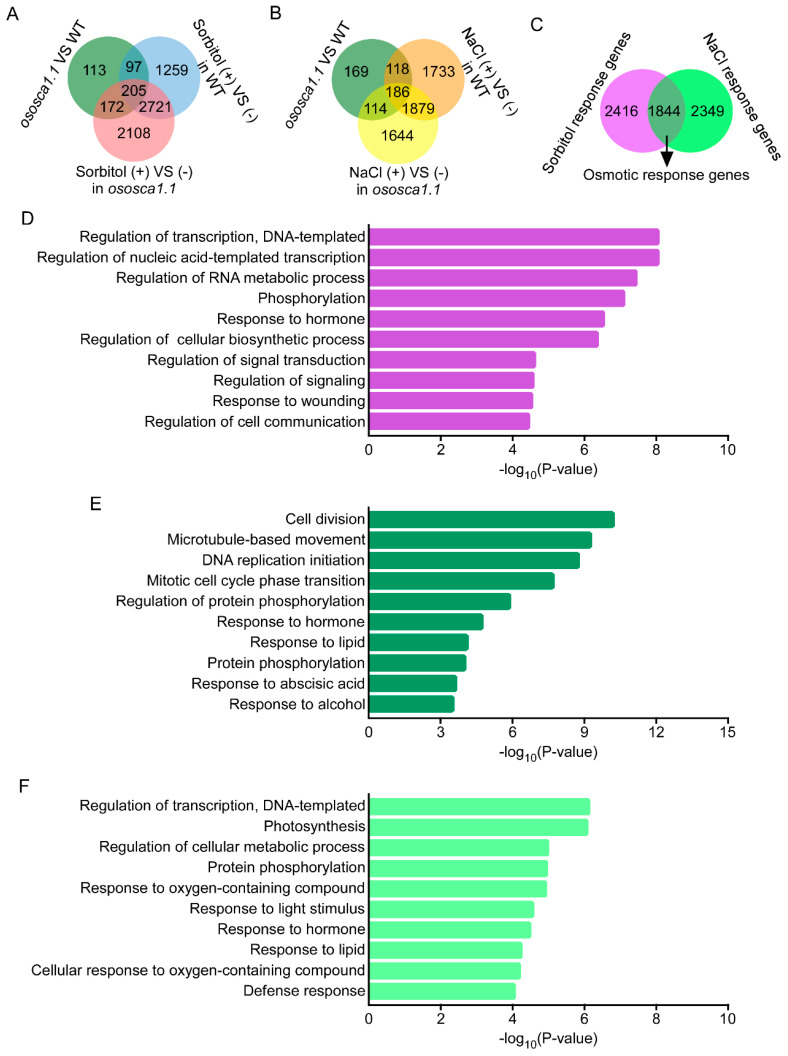
The enrichment analysis in the biological process category of Gene Ontology (GO) with OsOSCA1.1-regulated genes in 30-day-old rice shoots in response to hyperosmolality and salt stress treatment. (**A**,**B**) Venn diagram showing unique and shared differentially expressed genes (DEGs) between ZH11 and *ososca1.1* plants treated with (**A**) 250 mM of sorbitol and (**B**) 125 mM of NaCl for 24 h. (**C**) Venn diagram showing OsOSCA1.1-regulated genes in response to hyperosmolality (250 mM of sorbitol) and salt stress (125 mM of NaCl) treatment. (**D**–**F**) Enrichment analysis in biological process category of GO, with OsOSCA1.1-regulated sorbitol response genes (**D**), osmotic stress response genes (**E**), and NaCl response genes (**F**). GO term information for DEGs was extracted from BLAST results against the SWISS-PROT database and the top 10 terms are displayed.

**Figure 4 biology-11-00678-f004:**
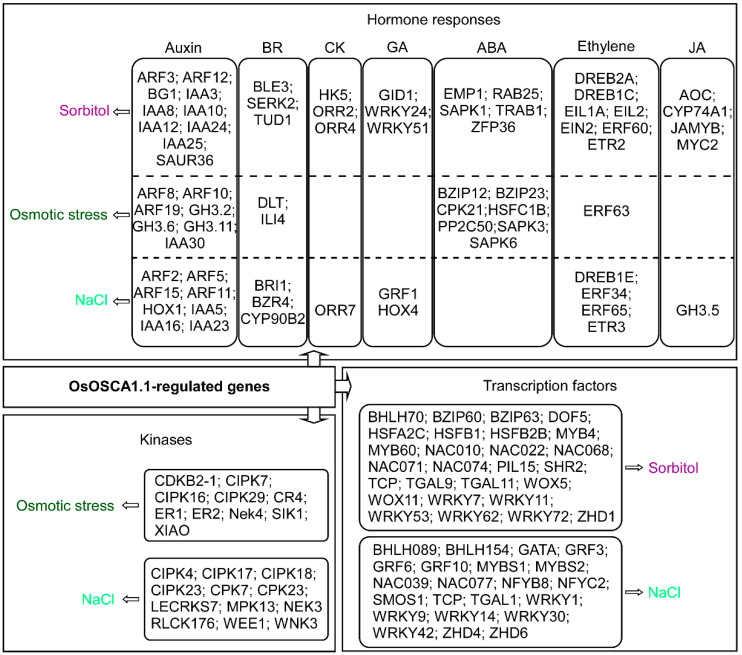
OsOSCA1.1-regulated genes enriched in hormone responses, gene transcriptional regulation, and kinases related to sorbitol stress (purple), osmotic stress (dark green), and NaCl stress (light green). The FPKM values of the listed genes are shown in Appendix A.

**Figure 5 biology-11-00678-f005:**
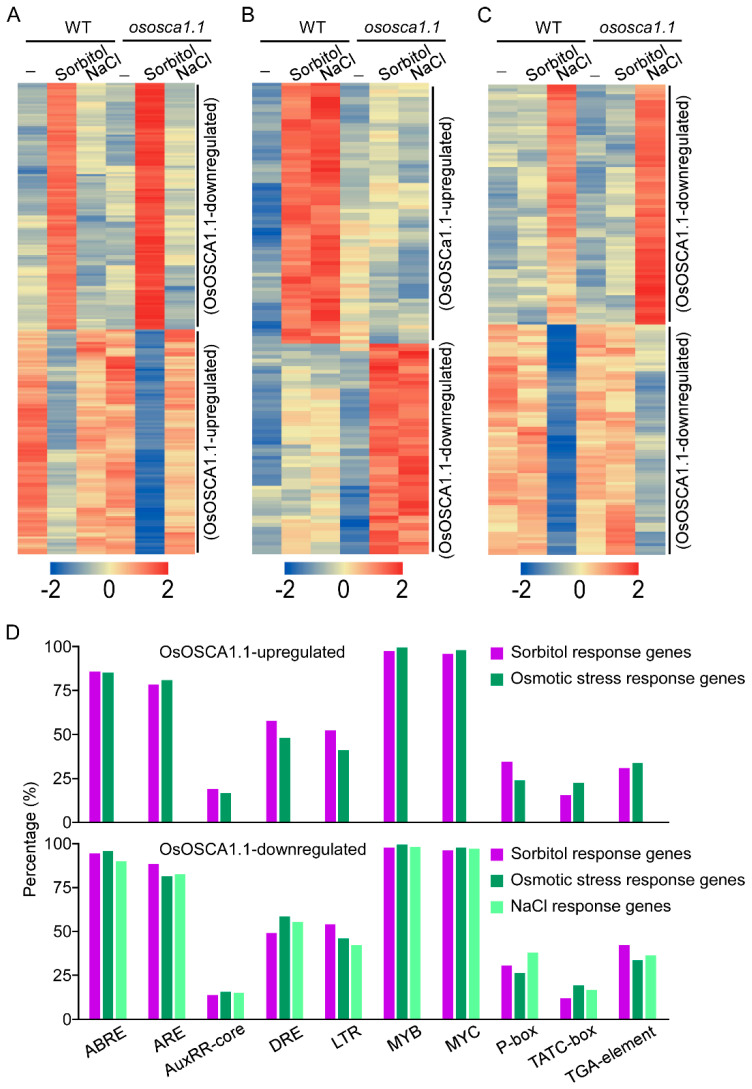
Enriched *cis*-regulatory elements in the 2000-bp promoter regions of three clustered OsOSCA1.1-regulated genes. (**A**–**C**) Hierarchical cluster analysis of the OsOSCA1.1-regulated genes in ZH11 and *ososca1.1* plants associated with (**A**) sorbitol, (**B**) osmotic stress, and (**C**) NaCl according to Cluster 3.0 software. In each cluster, one row represents one gene; log (FPKM) values of each gene were calculated, and expression patterns are color-coded; red and green indicate high and low expression, respectively. FPKM values for each gene in every cluster are listed in Appendix A. The gene expression pattern in each cluster was labeled on the right. (**D**) Different *cis*-regulatory elements were distributed in the 2000-bp promoter region of 178 OsOSCA1.1-upregulated genes (108 sorbitol response and 70 osmotic stress response), and 336 OsOSCA1.1-downregulated genes (119 sorbitol response, 56 osmotic stress response, and 126 NaCl response). The *cis*-regulatory elements were predicted using the PlantCARE program. The percentage of every cis-regulatory element distributed in the three types of OsOSCA1.1-regulated genes were calculated.

## Data Availability

The RNA-seq data were deposited in the National Center for Biotechnology Information (NCBI) Sequence Read Archive (SRA) (http://www.ncbi.blm.nih.gov/Traces/srahttp://www.ncbi.blm.nih.gov/Traces/sra, accessed on 27 May 2019), under the accession number PRJNA544967. All data generated or analyzed during this study are included in this article and its Appendix A.

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
