# Peer review of "OsOSCA1.1 Mediates Hyperosmolality and Salt Stress Sensing in Oryza sativa"

_biology, 2022, doi:10.3390/biology11050678_

Round 1

Reviewer 1 Report

The submitted manuscript entitled “OsOSCA1.1 mediates hyperosmolality and salt stress sensing 2 in Oryza sativa” presented functions of OsOSCA1.1, which is a reduced hyperosmolality-induced [Ca2+]i increase 1 (OSCA), in responses to osmotic and salinity stresses by analyses of ososca1.1 mutants. The authors concluded that OsOSCA1.1 mediates calcium signaling in root leading to stomatal closure and improvement of survival to those stresses. The authors also conducted transcriptome analysis and identified genes that differentially expressed between the ososca1.1 mutants and non-mutated plants under the stresses. I believe that this manuscript includes important knowledge about functions of the OsOSCA and is within the scope of Biology. However, I would ask the authors to address my concerns about this version before considering for publication.

I wonder whether the working hypothesis presented by the authors; OsOSCA1.1 mediates the expression of calcium signaling in root resulting in stomatal closure, is reasonable or not. I understood that the ososca1.1 mutants exhibited no increases of OICIcyt or SICIcyt in root treated with sorbitol or NaCl, and limited stomatal closure under the stresses. However, I think that disruption of OsOSCA1.1 could result in impaired calcium signaling in leaf as well, and it can simply explain less stomatal closure observed in ososca1.1 mutants exposed to stresses. In Arabidopsis, a mutant line for OSCA1 showed impaired osmotic calcium signaling in both guard cells and root cells (doi.org/10.1038/nature13593). I would ask the authors to consider the possibility; OsOSCA1.1 functions calcium signaling in leaf and controls movement of stomata.

The authors conducted RNA-seq using RNA extracted from shoots. Measurement of Ca2+ in shoots is desirable to link the gene expression analysis and the hypothesis of the function of OsOSCA1.1 in calcium signaling responding to stresses.

Minor points

Line 25: “ZH11” in abstract should be replaced by “Zhougha11”. Abbreviation without definition should be avoided in abstract.

Line 25: Increases of OICIcyt and SICIcyt in root of ososca1.1 mutants were smaller than WT (Fig. 1B and E). However, OICIcyt and SICIcyt did not reduce by the treatments. Please revise.

Line 299: “rhe” should be replace by “the”.

Results of semi-quantitative PCR shown in supplementary Fig. 1C seems like Western-blot. Please check.

Information of a file covers over Fig. 1E. Please remove.

Whereas the authors mainly focused on differentially expressed genes between ososca1.1 and WT under stress-conditions, they also identified 587 differentially expressed genes in the plants without stress-treatments. I would ask the authors to add some discussion regarding those genes that may be constitutively affected by OsOSCA1.1.

The authors used two types of osmolyte: sorbitol and PEG6000. I wonder whether we can consider the responses of rice plants to the osmotic stresses prepared by sorbitol and PEG are the same. I would ask the authors to provide some explanations about it.

Figure 5D is difficult to understand. Does “119” mean upregulated genes by sorbitol treatment? Formula for the percentage should be given instead of “Percent (%)”.

Author Response

Question 1 (Q1): I wonder whether the working hypothesis presented by the authors; OsOSCA1.1 mediates the expression of calcium signaling in root resulting in stomatal closure, is reasonable or not. I understood that the ososca1.1 mutants exhibited no increases of OICIcyt or SICIcyt in root treated with sorbitol or NaCl, and limited stomatal closure under the stresses. However, I think that disruption of OsOSCA1.1 could result in impaired calcium signaling in leaf as well, and it can simply explain less stomatal closure observed in ososca1.1 mutants exposed to stresses. In Arabidopsis, a mutant line for OSCA1 showed impaired osmotic calcium signaling in both guard cells and root cells (doi.org/10.1038/nature13593). I would ask the authors to consider the possibility; OsOSCA1.1 functions calcium signaling in leaf and controls movement of stomata.

Answer 1 (A1): We thank the reviewer’s very good comments about OsOSCA1.1 mediated calcium signaling in rice leaves. In this study, we monitored the intracellular calcium change with Yellow Cameleon YC 3.6 in rice roots, which is difficult to use for calcium measurement in rice leaves because of their specific structure. This is the reason for our measuring the stomatal movement of rice leaves with an environmental scanning electron microscope in this study. In addition, we can use another calcium indicator, aequorin, to measure intracellular calcium change in rice leaves. This is our ongoing project in rice, which need a long time to get the results because we need more time to obtain the stable-genetically transgenic rice.

Q2: The authors conducted RNA-seq using RNA extracted from shoots. Measurement of Ca2+ in shoots is desirable to link the gene expression analysis and the hypothesis of the function of OsOSCA1.1 in calcium signaling responding to stresses.

A2: We acknowledge the reviewer’s comments and give the following explanations: In this study, we measured the calcium change in roots, stomatal movement, and survival rate after the stress treatments on rice roots. Because of the transpiration in leaves, the stress signals can transduce from the roots to leaves, then induce stomatal closure and the change of gene expression. Therefore, we analyze the gene expression in leaves after the stress treatment on rice roots.

Q3: Line 25: “ZH11” in abstract should be replaced by “Zhougha11”. Abbreviation without definition should be avoided in abstract.

A3: We thank the reviewer’s comment and replace “ZH11” with “Zhonghua11” in the abstract (Page 1, Line 41).

Q4: Line 25: Increases of OICIcyt and SICIcyt in root of ososca1.1 mutants were smaller than WT (Fig. 1B and E). However, OICIcyt and SICIcyt did not reduce by the treatments. Please revise.

A4: We acknowledge the reviewer’s comments and revise the sentence in the abstract and result section.

Q5: Line 299: “rhe” should be replace by “the”.

A5: We thank the reviewer’s comments and revise the sentence in the text (Page 12, Line 418).

Q6: Results of semi-quantitative PCR shown in supplementary Fig. 1C seems like Western-blot. Please check.

A6: We acknowledge the reviewer’s comments and replace the Supplementary Figure 1C with qRT-PCR results.

Q7: Information of a file covers over Fig. 1E. Please remove.

A7: We thank the reviewer’s comments and remove it in a new version.

Q8: Whereas the authors mainly focused on differentially expressed genes between ososca1.1 and WT under stress-conditions, they also identified 587 differentially expressed genes in the plants without stress-treatments. I would ask the authors to add some discussion regarding those genes that may be constitutively affected by OsOSCA1.1.

A8: We acknowledge the reviewer’s comments and give the following explanations: Although we identified 587 DEGs between WT and ososca1.1 plants, most of them are still induced or repressed by hyperosmolality and salt stress treatments in this study. Therefore, we focused on OsOSCA1.1-regulated genes in response to environmental stresses in this study.

Q9: The authors used two types of osmolyte: sorbitol and PEG6000. I wonder whether we can consider the responses of rice plants to the osmotic stresses prepared by sorbitol and PEG are the same. I would ask the authors to provide some explanations about it.

A9: We thank the reviewer’s comments and give the following explanations: Firstly, sorbitol and PEG 6000 are two different osmolytes: sorbitol is cell permeable, but PEG 6000 is cell impermeable. However, both of them can be used for monitoring drought stress treatment because they can produce the osmotic stress signal. Therefore, in this study, we used them for the treatment of rice roots as hyperosmolarity stresses. In addition, we herein found an interesting thing: With the transcriptomic analysis, we characterized three types of OsOSCA1.1-regulated genes: sorbitol response genes, osmotic stress response genes and NaCl response genes. Therefore, we predicted that there existed the PEG response genes if we treated the rice roots with PEG 6000.

Q10: Figure 5D is difficult to understand. Does “119” mean upregulated genes by sorbitol treatment? Formula for the percentage should be given instead of “Percent (%)”.

A10: We acknowledge the reviewer’s comments and reorganize the data in Figure 5.

Reviewer 2 Report

This research expanded on the authors' previous work on the functional characterization of the rice OSCA gene family. Using a functional genomic approach, the authors aimed to characterize the OsOSCA1.1 gene in rice. Using both mutant and functional complementation techniques, the authors demonstrated that this gene appears to be critical for stomatal closure, root development, and survival to both hyperosmotic and salt stresses. Transcriptome analysis revealed the genes involved in these stress responses in both the Zhonghua11 cultivar and the ososca1.1 mutant. A cis-regulatory element analysis was performed on the potentially co-regulated genes.

The paper is mostly well and well-presented. It contributes to the understanding of the potential function(s) of this rice OSCA gene member. Because salinity and hyperosmolarity are two major stresses, discovering key genes implicated in these responses would be of significant importance for future breading development.

Regardless of the quality of the present work, I have some comments to attempt to improve it more.

  • Figure 1E: Delete the box that was placed.

  • For publication, all RNAseq datasets should be deposited to GEO or any other publicly available depository.

  • Some arguments in paragraphs 3.4 and 3.5 are discussion, and should be replaced in this section. In particular, in paragraph 3.5, there are far more discussion aspects regarding cis-regulatory elements than in the discussion paragraph itself. ABRE elements, for example, are key ABA-responsive elements that, I agree, may be linked as well to calcium signaling via ABA signal transduction. In the Discussion section, it is less evident than in this paragraph.

  • - As with cis-acting elements, the authors have a tendency to use shortcuts to support their hypotheses (in the discussion section, ABRE = calcium signaling rather than ABA signaling). The main, problematic one, is the distinction between association and direct regulation. With their results, the authors can only suggest several genes related with salt and osmotic stressors and the deregulation of 1 gene expression using their experimental RNAseq approach, but they cannot rule out other actors (phytohormones, for example) or other OSACA members' functions. Indeed, as pointed by the authors in their previous work: “These results indicated that multiple members of the OsOSCA family have redundant functions in osmotic sensing and diverse roles in stress adaption”. So, please be less definitive; here, you only evidenced some promising associations, but more research is required to prove a direct relationship/regulation by OSCA1.1.

  • Section 3.5 title makes nonsense: “Known cis-regulatory elements are overrepresented in the promoters of OsOSCA1.1-regulatedgenes”. Cis-acting elements are present in all promoter regions, regardless of their type. It would be informative to indicate which kind of "known cis-regulatory components" are overrepresented.

  • Please explain how you came at the hypothesis that these cis-regulatory elements were overrepresented. It is not obvious in the Materials and Methods. We can only deduce from this M&M description that you analyzed the presence versus absence of certain regulatory elements but did not really determine their overrepresentation, because you did not compare with “non-induced” genes.

Author Response

Question 1 (Q1): Figure 1E: Delete the box that was placed.

Answer 1 (A1): We thank the reviewer’s comments and remove the box in a new version of Figure 1.

Q2: For publication, all RNAseq datasets should be deposited to GEO or any other publicly available depository.

A2: We acknowledge the reviewer’s comments and deposit RNAseq data in NCBI Sequence Read Archive (SRA) (http://www.ncbi.blm.nih.gov/Traces/srahttp://www.ncbi.blm.nih.gov/Traces/sra) under accession number PRJNA544967 in Data Availability (Page 13, Line 497-501).

Q3: Some arguments in paragraphs 3.4 and 3.5 are discussion, and should be replaced in this section. In particular, in paragraph 3.5, there are far more discussion aspects regarding cis-regulatory elements than in the discussion paragraph itself. ABRE elements, for example, are key ABA-responsive elements that, I agree, may be linked as well to calcium signaling via ABA signal transduction. In the Discussion section, it is less evident than in this paragraph.

A3: We thank the reviewer’s comments and revise the paragraphs 3.4 and 3.5 in the text.

Q4: - As with cis-acting elements, the authors have a tendency to use shortcuts to support their hypotheses (in the discussion section, ABRE = calcium signaling rather than ABA signaling). The main, problematic one, is the distinction between association and direct regulation. With their results, the authors can only suggest several genes related with salt and osmotic stressors and the deregulation of 1 gene expression using their experimental RNAseq approach, but they cannot rule out other actors (phytohormones, for example) or other OSACA members' functions. Indeed, as pointed by the authors in their previous work: “These results indicated that multiple members of the OsOSCA family have redundant functions in osmotic sensing and diverse roles in stress adaption”. So, please be less definitive; here, you only evidenced some promising associations, but more research is required to prove a direct relationship/regulation by OSCA1.1.

A4: We acknowledge the reviewer’s comments and rewrite the sentences in the text (Page 12, Line 423-438)

Q5: Section 3.5 title makes nonsense: “Known cis-regulatory elements are overrepresented in the promoters of OsOSCA1.1-regulatedgenes”. Cis-acting elements are present in all promoter regions, regardless of their type. It would be informative to indicate which kind of "known cis-regulatory components" are overrepresented.

A5: We thank the reviewer’s comments and give the detail cis-regulatory site in the Section 3.5 title.

Q6: Please explain how you came at the hypothesis that these cis-regulatory elements were overrepresented. It is not obvious in the Materials and Methods. We can only deduce from this M&M description that you analyzed the presence versus absence of certain regulatory elements but did not really determine their overrepresentation, because you did not compare with “non-induced” genes.

A6: We thank the reviewer’s comments and revise the contents in the Result and Discussion section.

Reviewer 3 Report

Biology-MDPI

MS No.: biology-1690018

Manuscript Title: "OsOSCA1.1 mediates hyperosmolality and salt stress sensing in Oryza sativa"

Reviewer comments:

The current study has significant value to improve the salt stress sensing in Oryza sativa by OsOSCA1.1. However, the manuscript requires improvement for readability and understanding. Please see my comments below. The topic fits well with the scope of Biology-MDPI and the results are of interest to the scientific community. This study is well-designed and the methods are satisfy. However, the manuscript needs a major revision before publication. In addition, the innovation is insufficient and some of the discussion is inadequate. It will be deserved a minor revision before consideration of publication in Biology-MDPI

Abstract:

  • Lines 30 and 33: “Transcriptomic analysis further revealed three groups of OsOSCA1.1-regulated genes in the shoots, along with 2,416 sorbitol-responsive genes, 2,349 NaCl-responsive genes, and 1,844 common osmotic stress-responsive genes following treatment with 250 mM sorbitol and 125 mM NaCl in 30-day-old rice roots for 24 h”. Please paraphrase this sentence and connect it with the previous and next sentences.
  • Lines 37-38:” These results indicate that OsOSCA-mediated calcium signalling regulates gene expression in response to drought and salt stress in rice”. It is a good summary and general conclusion of this research but although there is a strong association between salinity and drought, the authors should focus on salt stress only.
  • Lines 26-29:” Hyperosmolality- and salt stress-induced stomatal closure were also disrupted in leaves of 30-day-old ososca1.1 mutants, resulting in lower stomatal resistance and survival rates than observed in ZH11. However, overexpression of OsOSCA1.1 in ososca1.1 complemented these growth phenotypes related to hyperosmolality and salt stress”. These sentences constitute a broad meaning, please be specific by including specialized words that fit the topic of the manuscript.
  • Authors should write the best treatments that gave the highest results as a recommendation in the penultimate paragraph in the abstract section.
  • The abbreviation should be used after the full term. Please be consistent with the usage of all abbreviations. Pls revise the abbreviations in the whole part of MS.

Keywords: “OsOSCA; OICIcyt; SICIcyt” It is preferable not to write abbreviations in keywords, especially since these abbreviations are written for the first time. In addition, I suggest rephrasing some words because keywords should not repeat words from the title.

Introduction:

  • The current state of MS needs some attention.
  • Lines 42-44: “As sessile organisms, plants are exposed to continual abiotic stress, including drought and salt stress, throughout their growth period, i.e. from seed germination to flowering [1-3]”. Pls be specific and talk about salt stress only.
  • Lines 55-61: “These key genes are transcriptionally regulated by several families of ranscription factors (TFs), including ABA-responsive element-binding protein (AREB)/ABRE binding factor (ABF), dehydration-responsive element-binding protein (DREB), ethylene-responsive factor (ERF), no apical meristem (NAM), ATAF1/2, and cup-58shaped cotyledon (CUC2) (NAC) TFs. The basic leucine zipper (bZIP) family proteins MYC and MYB modulate the expression of stress-related genes by binding to cis-regula-60tory elements in the promoter region [11-13]”. Pls rephrase and shorten.
  • Lines 49 and 50: “Thus, drought and salt stress have both distinct 49and overlapping features”. Pls add relevant ref.
  • Lines 55-59:” These key genes are transcriptionally regulated by several families of transcription factors (TFs), including ABA-responsive element-binding protein (AREB)/ABRE binding factor (ABF), dehydration-responsive element-binding protein (DREB), ethylene-responsive factor (ERF), no apical meristem (NAM), ATAF1/2, and cupshaped cotyledon (CUC2) (NAC) TFs”. Pls add relevant ref.
  • The plagiarism should be reduced according to the journal's requirements.
  • Lines 49 and 50, and 61 and 62: pls conjunction these two following sentences. Please connect the following two sentences. “Thus, drought and salt stress have both distinct and overlapping features” and “However, the regulatory mechanism that allows plants to distinguish drought and salt stress signals remains unknown”.
  • Pls add a short paragraph about the importance and novelty of the study compared to previous studies in this regard.

Materials and methods:

  • The materials and methods section is well written but relevant references have to add to support them.
  • The authors did not mention anything about the design of the experiment, the number of transactions, nor the number of replications in the materials and methods part, neither at the beginning of this part nor at the end (in the statistical analysis part). This is a critical point.
  • It is important to add relevant and recent references regarding all methods used in this section.
  • Pls shorten the materials and methods section; there are more unnecessary details in several parts.

Results and discussion

  • Figure 1. Pls the text box from the shape to the right until what is written below it appears.
  • Figure 3. Gene Ontology (Go) terms should be distributed into three categories:
  1. Biological Processes
  2. Cell components
  3. Molecular functions
  • Figure 3. The vertical axis has to label
  • The authors used references in the results part, this should be in the discussion part, so this part should be organized normally.
  • Please highlight more specifically the objective of the work.
  • The results part is very long, this part should be shortened as much as possible in order to reach the intended meaning directly for all the studied characteristics
  • In the discussion section, conjunctions should be used to show the relationship between sentences.
  • Lines 356-359:” We found that, in rice roots, OsOSCA1.1 mediates OICIcyt and SICIcyt, which are essential for stomatal closure and seedling survival in response to hyperosmolality and salt stress treatment. Therefore, we demonstrated that, similar to OSCA1 in Arabidopsis, OsOSCA1.1 functions as an osmosensor in rice:” pls avoid using the personal pronouns in the entire manuscript.
  • What do you mean by “Functional annotation of these DEGs showed that most encoded multiple members of different signaling networks, including hormone- and calcium-signaling pathways, mitogen-activated protein kinase (MAPK) cascades, and TFs; these likely play important roles in the drought and salt stress responses of rice.
  • In many places in the entire manuscript, especially the discussion section, there is talk about drought and salinity, but what should be focused on is salt stress despite the strong correlation between them as mentioned previously.
  • The resolution of all figures was not clear enough, so pls pay attention to this comment.
  • Please, make an effort to synthesize the text avoiding redundancies and repetitions in the discussion.
  • Discussion in several parts is confusing, suggesting rewriting.
  • Some parts of the discussion sentences need clarification and interpretation, and recent references need to be used as much as possible.
  • The discussion should be better organized. It is important to try to better deepen and explain.

Conclusions; the authors should write a summary of your work in short sentences so that I, as a reader of this article, can understand what the article ended up being.

References

  • The number of references is about 45 ref. I think it is not satisfy. Seven of them during the last five years. So, pls delete the old ones and avoid repetition. There is a recent ref. (2020-2022) in the same trend of the topic of this MS, pls pay attention to this point and cross-check all the references for mistakes, and follow the journal style of reference input.

General comments:

  • To summarize, this is a good study, which certainly merits a publication after a minor revision.
  • The manuscript contains some typo errors; please revise it very carefully. A careful revision of the English Grammar is required. So, language needs to be improved thoroughly

Author Response

Question 1 (Q1): Lines 30 and 33: “Transcriptomic analysis further revealed three groups of OsOSCA1.1-regulated genes in the shoots, along with 2,416 sorbitol-responsive genes, 2,349 NaCl-responsive genes, and 1,844 common osmotic stress-responsive genes following treatment with 250 mM sorbitol and 125 mM NaCl in 30-day-old rice roots for 24 h”. Please paraphrase this sentence and connect it with the previous and next sentences.

Answer 1 (A1): We thank the reviewer’s comments and rephrase the sentence in the text (Page 1, Line 44-47).

Q2: Lines 37-38:” These results indicate that OsOSCA-mediated calcium signalling regulates gene expression in response to drought and salt stress in rice”. It is a good summary and general conclusion of this research but although there is a strong association between salinity and drought, the authors should focus on salt stress only.

A2: We acknowledge the reviewer’s comments and give the following explanations: In this study, we found that OsOSCA mediated both hyperosmolality induced calcium increases and salt stress induced calcium increases, which is essential for hyperosmolality induced stomatal closure and salt induced stomatal closure, respectively. Therefore, we focus on both drought and salt stresses in our works.

Q3: Lines 26-29:” Hyperosmolality- and salt stress-induced stomatal closure were also disrupted in leaves of 30-day-old ososca1.1 mutants, resulting in lower stomatal resistance and survival rates than observed in ZH11. However, overexpression of OsOSCA1.1 in ososca1.1 complemented these growth phenotypes related to hyperosmolality and salt stress”. These sentences constitute a broad meaning, please be specific by including specialized words that fit the topic of the manuscript.

A3: We thank the reviewer’s comments and rephrase the sentence in the text (Page 1, Line 40-44).

Q4: Authors should write the best treatments that gave the highest results as a recommendation in the penultimate paragraph in the abstract section.

A4: We acknowledge the reviewer’s comments and give the next explanations: In this study, we just use the appropriate concentration of sorbitol and NaCl as the hyperosmolality and salt stress treatment, respectively, which trigger the [Ca2+]cyt increases in rice roots, and associated stomatal closure and gene expression in shoots. However, we do not know the best treatment for the rice.

Q5: The abbreviation should be used after the full term. Please be consistent with the usage of all abbreviations. Pls revise the abbreviations in the whole part of MS.

A5: We thank the reviewer’s comments and check the all abbreviation in this manuscript.

Q6: Keywords: “OsOSCA; OICIcyt; SICIcyt” It is preferable not to write abbreviations in keywords, especially since these abbreviations are written for the first time. In addition, I suggest rephrasing some words because keywords should not repeat words from the title.

A6: We acknowledge the reviewer’s comments and give the full name in the Keyword.

Q7: Lines 42-44: “As sessile organisms, plants are exposed to continual abiotic stress, including drought and salt stress, throughout their growth period, i.e. from seed germination to flowering [1-3]”. Pls be specific and talk about salt stress only.

A7: We thank the reviewer’s comments and give the next explanations: In this study, we revealed the biological function of OsOSCA1.1 on mediating drought and salt induced calcium increases, but not salt stress only.

Q8: Lines 55-61: “These key genes are transcriptionally regulated by several families of ranscription factors (TFs), including ABA-responsive element-binding protein (AREB)/ABRE binding factor (ABF), dehydration-responsive element-binding protein (DREB), ethylene-responsive factor (ERF), no apical meristem (NAM), ATAF1/2, and cup-58shaped cotyledon (CUC2) (NAC) TFs. The basic leucine zipper (bZIP) family proteins MYC and MYB modulate the expression of stress-related genes by binding to cis-regulatory elements in the promoter region [11-13]”. Pls rephrase and shorten.

A8: We acknowledge the reviewer’s comments and rephrase the sentence in the text (Page 2, Line 67-72).

Q9: Lines 49 and 50: “Thus, drought and salt stress have both distinct and overlapping features”. Pls add relevant ref.

A9: We thank the reviewer’s comments and add the reference in the text.

Q10: Lines 55-59:” These key genes are transcriptionally regulated by several families of transcription factors (TFs), including ABA-responsive element-binding protein (AREB)/ABRE binding factor (ABF), dehydration-responsive element-binding protein (DREB), ethylene-responsive factor (ERF), no apical meristem (NAM), ATAF1/2, and cupshaped cotyledon (CUC2) (NAC) TFs”. Pls add relevant ref.

A10: We acknowledge the reviewer’s comments and add the reference in the text.

Q11: The plagiarism should be reduced according to the journal's requirements.

A11: We don’t agree with reviewer’s comments on plagiarism and give the following explanations: Because we recently published three papers about OsOSCAs [Li, BMC Plant Biology (2015) 15: 261; Zhai, Cell Calcium (2020) 91: 102261 and Zhai, Transgenic Research (2021) 30: 811-820], the words and terms about OsOSCAs in Introduction, Material method and Results are inevitably repeated. However, we don’t think this kind of repetitions are plagiarism.

Q12: Lines 49 and 50, and 61 and 62: pls conjunction these two following sentences. Please connect the following two sentences. “Thus, drought and salt stress have both distinct and overlapping features” and “However, the regulatory mechanism that allows plants to distinguish drought and salt stress signals remains unknown”.

Pls add a short paragraph about the importance and novelty of the study compared to previous studies in this regard.

A12: We acknowledge the reviewer’s comments and rephrase the sentence in the text.

Q13: The materials and methods section is well written but relevant references have to add to support them.

A13: We thank the reviewer’s comments and add some reference in materials and method section.

Q14: The authors did not mention anything about the design of the experiment, the number of transactions, nor the number of replications in the materials and methods part, neither at the beginning of this part nor at the end (in the statistical analysis part). This is a critical point.

A14: We acknowledge the reviewer’s comments and give the detail experimental replications and statistical analyses in every figure legend if needed.

Q15: It is important to add relevant and recent references regarding all methods used in this section. Pls shorten the materials and methods section; there are more unnecessary details in several parts.

A15: We thank the reviewer’s comments, and revise the method section and add some relevant references in the text.

Q16: Figure 1. Pls the text box from the shape to the right until what is written below it appears.

A16: We acknowledge the reviewer’s comments and give a new version of Figure 1.

Q17: Figure 3. Gene Ontology (Go) terms should be distributed into three categories: Biological Processes, Cell components,Molecular functions

A17: We thank the reviewer’s comments and add the data of other two categories in the results (Supplementary Figure 4 and 5).

Q18: Figure 3. The vertical axis has to label

A18: We acknowledge the reviewer’s comments and present the various terms of GO Biological Process category with vertical axis in Figure 3D.

Q19: The authors used references in the results part, this should be in the discussion part, so this part should be organized normally.

A19: We thank the reviewer’s comments, and remove the references from the result section and discuss the content in the Discuss section (Page 12, Line 427-435)

Q20: Please highlight more specifically the objective of the work.

A20: We acknowledge the reviewer’s comments and give a simple summary in the text.

Q21: The results part is very long, this part should be shortened as much as possible in order to reach the intended meaning directly for all the studied characteristics. In the discussion section, conjunctions should be used to show the relationship between sentences.

A21: We thank the reviewer’s comments and carefully revise the result and discuss section in the text.

Q22: Lines 356-359:” We found that, in rice roots, OsOSCA1.1 mediates OICIcyt and SICIcyt, which are essential for stomatal closure and seedling survival in response to hyperosmolality and salt stress treatment. Therefore, we demonstrated that, similar to OSCA1 in Arabidopsis, OsOSCA1.1 functions as an osmosensor in rice:” pls avoid using the personal pronouns in the entire manuscript.

A22: We acknowledge the reviewer’s comments and revise the sentence in the text (Page 10, Line 369-377).

Q23: What do you mean by “Functional annotation of these DEGs showed that most encoded multiple members of different signaling networks, including hormone- and calcium-signaling pathways, mitogen-activated protein kinase (MAPK) cascades, and TFs; these likely play important roles in the drought and salt stress responses of rice.

A23: We thank the reviewer’s comments and revise this sentence in the text (Page 12, Line 394-398)

Q24: In many places in the entire manuscript, especially the discussion section, there is talk about drought and salinity, but what should be focused on is salt stress despite the strong correlation between them as mentioned previously.

A24: We acknowledge the reviewer’s comments and give the next explanations: In this study, we revealed the biological function of OsOSCA1.1 on mediating drought and salt induced calcium increases, but not salt stress only.

Q25: The resolution of all figures was not clear enough, so pls pay attention to this comment.

A25: We thank the reviewer’s comments and give the high-resolution figures in the text.

Q26: Please, make an effort to synthesize the text avoiding redundancies and repetitions in the discussion.

Discussion in several parts is confusing, suggesting rewriting.

Some parts of the discussion sentences need clarification and interpretation, and recent references need to be used as much as possible.

The discussion should be better organized. It is important to try to better deepen and explain.

A26: We acknowledge the reviewer’s comments, and revise the sentence in the Discuss section, remove some old reference and present some new reference.

Q27: Conclusions; the authors should write a summary of your work in short sentences so that I, as a reader of this article, can understand what the article ended up being.

A27: We thank the reviewer’s comments and revise the conclusion section in this text.

Q28: The number of references is about 45 ref. I think it is not satisfy. Seven of them during the last five years. So, pls delete the old ones and avoid repetition. There is a recent ref. (2020-2022) in the same trend of the topic of this MS, pls pay attention to this point and cross-check all the references for mistakes, and follow the journal style of reference input.

A28: We acknowledge the reviewer’s comments and update the reference in our manuscript.

Q29: The manuscript contains some typo errors; please revise it very carefully. A careful revision of the English Grammar is required. So, language needs to be improved thoroughly

A29: We thank the reviewer’s comments and thoroughly revise the manuscript. In addition, we invite two professional editors, both native speakers of English, to check the English in this manuscript.

Round 2

Reviewer 1 Report

I believe that the resubmitted manuscript entitled “OsOSCA1.1 mediates hyperosmolality and salt stress sensing in Oryza sativa” has been well-revised. I would thank for sincere responses to the reviewer's comments. However, the authors did not address several points that I had raised.

The authors politely explained about relationship between OSCIcyt or SICIcyt in root and stomatal closure in the letter to the reviewer. However, no such descriptions were added to the manuscript. I am still wondering whether less stomatal closure observed in ososca1.1 mutants was caused by impaired stress signaling from root to leaf or by impaired calcium signaling in guard cells, or both. I knew that the authors do not have any evidences for the behavior of calcium ion in guard cells so far; however, I would like to ask the authors to show their opinions with relevant references.

Line 39: Let me repeat it, increases of OICIcyt and SICIcyt in root of ososca1.1 mutants were smaller than WT (Fig. 1B and E); however, OICIcyt and SICIcyt did not reduce by the treatments. I would ask the authors to consider “increased only slightly” or something like that instead of “were reduced”.

The authors described that the most of 587 differentially expressed genes in plants without stress-treatment were stress-inducible or stress-repressive in the letter. This fact may show that OsOSCA1.1 is a key gene in stress responses of rice plants. I would ask authors to put a sentence showing this fact: most of the 587 DEGs were stress-inducible or stress-repressive, in the manuscript.

Fig. 5D: Why did the bars for NaCl response genes disappear?

Author Response

Response to reviewer’s comments

Question 1 (Q1): The authors politely explained about relationship between OSCIcyt or SICIcyt in root and stomatal closure in the letter to the reviewer. However, no such descriptions were added to the manuscript. I am still wondering whether less stomatal closure observed in ososca1.1 mutants was caused by impaired stress signaling from root to leaf or by impaired calcium signaling in guard cells, or both. I knew that the authors do not have any evidences for the behavior of calcium ion in guard cells so far; however, I would like to ask the authors to show their opinions with relevant references.

Answer 1 (A1): We acknowledge the reviewer’s comments and give the following explanations: Firstly, we replace the word “essential for” with “associated with”, and “promotes” with “is related with” in the text (Page 1, Line 22; Page 7, Line 251; and Page 10, Line 372), which suggested that the abolished OSCIcyt or SICIcyt in root cells are related with the disrupted hyperosmolality-induced stomatal closure and salt stress-induced stomatal closure in ososca1.1 plants. Secondly, in our previously published paper (Science, 2007, 315: 1423-1427), we found that Arabidopsis wild-type plant displayed diurnal oscillations in 45Ca uptake and transport to the leaves from the roots, which confirm previous reports that Ca2+ uptake and transport to leaves are controlled mainly by transpiration that is regulated by stomatal conductance (Plant Cell, 2005, 17, 2142 and Ann Bot, 2003, 92, 487). These results suggested that osmolytes or ion can transport to leaves from roots by transpiration. On the other hand, Choi et al (Annu Rev Plant Biol,. 2016, 67: 287-307) reviewed that a rapid, long-distance calcium signaling happened in plants. Therefore, according to our results, we conclude that OsOSCA1.1 mediates OICIcyt and SICIcyt in rice roots, which are associated with stomatal closure and seedling survival in response to hyperosmolality and salt stress treatment in our manuscript. However, because the above reference is not directly associated with our results, we have not cited them in our manuscript.

Q2: Line 39: Let me repeat it, increases of OICIcyt and SICIcyt in root of ososca1.1 mutants were smaller than WT (Fig. 1B and E); however, OICIcyt and SICIcyt did not reduce by the treatments. I would ask the authors to consider “increased only slightly” or something like that instead of “were reduced”.

A2: We thank the reviewer’s comments and replace the word “reduced” with “abolished” in the text.

Q3: The authors described that the most of 587 differentially expressed genes in plants without stress-treatment were stress-inducible or stress-repressive in the letter. This fact may show that OsOSCA1.1 is a key gene in stress responses of rice plants. I would ask authors to put a sentence showing this fact: most of the 587 DEGs were stress-inducible or stress-repressive, in the manuscript.

A3: We acknowledge the reviewer’s comments and rephrase the results about these 587 DEGs in the text (Page 7, Line 265-271).

Q4: Fig. 5D: Why did the bars for NaCl response genes disappear?

A4: We thank the reviewer’s comments and give the next explanations: Because NaCl response genes we presented in Figure 5C are OsOSCA1.1-downregulated, we only showed OsOSCA1.1-upregulated sorbitol- and osmotic stress response genes in the top of Figure 5D.